# LFNO: Bridging Laplace and Fourier for Effective Operator Learning

## Abstract

We introduce Laplace–Fourier Neural Operator (LFNO), a novel operator learning model that bridges the strengths of Laplace Neural Operators (LNO) and Fourier Neural Operators (FNO). By combining the transient response of LNO with the steady-state response of FNO through the Fourier integral operator, our model enables capturing transient behavior more effectively than both LNO and FNO while remaining comparable on linear and nonlinear PDEs. We demonstrate LFNO's effectiveness on solving three ODEs (Duffing, Lorenz, Pendulum) and five PDEs (Euler-Bernoulli beam, diffusion, reaction-diffusion, Brusselator, Gray-Scott) in comparison to FNO and LNO. These results highlight LFNO's ability to unify transient and steady-state modeling, delivering superior accuracy and stability across various dynamical systems.

## 1 Introduction

In recent years, advancements through machine learning have emerged as a powerful paradigm for solving differential equations in several applications, such as fluid dynamics (Lomax et al., 2001; Brunton et al., 2016; Ling et al., 2016; Tartakovsky et al., 2020; Ranade et al., 2021), electromagnetism (Xiong et al., 2019; Baldan et al., 2021; Zhang et al., 2021), and quantum mechanics (Brockherde et al., 2017; Yu et al., 2018; Hermann et al., 2020). However, many of these approaches still struggle to handle problems with diverse boundary conditions or high-dimensional dynamics efficiently.

Neural operator frameworks were proposed to overcome these obstacles by directly learning the mapping between input functions and solutions. (Lu et al., 2019; 2021) represented input functions through a branch network and query locations through a trunk network, enabling the approximation of highly nonlinear operators across diverse domains. Based on this concept, neural networks for solving partial differential equations (PDEs), such as those in (Li et al., 2020b;c), emerged to generalize resolution-invariant methods. Fourier Neural Operator (FNO) (Li et al., 2020a) was later introduced as a spectral approach to operator learning. FNO uses Fourier transforms to parameterize integral kernels in the frequency domain, retaining and updating only low-frequency modes for resolution invariance, efficient training, and strong performance on large-scale PDE benchmarks. (Shin et al., 2022) theoretically and experimentally demonstrated that extending the FNO through the refinement of Fourier-based kernels can enhance expressiveness in high-frequency or inhomogeneous scenarios. However, its reliance on periodic boundary conditions and regular grids limits its applicability to general geometries. While some studies have attempted to address these limitations by incorporating additional techniques into existing models (Lin and Chen, 2023; Li et al., 2024; Diab and Al-Kobaisi, 2025; Eshaghi et al., 2025), other works have aimed to solve the problem by expanding the domain.

(Fanaskov and Oseledets, 2023) enhances representation by combining Fourier series with Chebyshev bases, improving the approximation beyond periodic settings. (Bonev et al., 2023) extends the FNO approach to respect the geometry of the two-dimensional sphere, which applies to forecast atmospheric dynamics, and (Li et al., 2023; Tran et al., 2023) generalizes neural operators from uniform rectangular meshes to arbitrary geometries, expanding their applicability to complex spatial domains. More recently, Laplace Neural Operator (LNO) (Cao et al., 2024) was proposed to extend operator learning to the Laplace domain. By exploiting a pole–residue formulation, LNO naturally

captures both transient and steady-state behaviors of dynamical systems, offering improved accuracy for problems with non-periodic or exponentially decaying responses.

Here, inspired by the formulation of LNO (Cao et al., 2024), we propose a unified framework, Laplace–Fourier Neural Operator (LFNO) that combines the strengths of LNO and FNO. Our approach leverages LNO to separate transient and steady-state responses, and applies the Fourier integral operator to model the steady-state component effectively in the frequency domain. Our contributions are as follows.

- We propose LFNO, which unifies the theoretical approaches of LNO and FNO.
- Our approach shows that empirically effective operator learning can be achieved through this integration.
- We demonstrate that LFNO outperforms both LNO and FNO on ordinary differential equations (ODEs) with transient behavior while achieving performance comparable to that of FNO on general PDEs.

## 2 METHODOLOGY

This section details how the complementary strengths of the LNO and FNO are combined to develop our proposed method.

In (Cao et al., 2024), LNO models the dynamic response of a system using the Laplace transform and the pole-residue formulation, and its output is divided into a transient response that decays over time and a steady-state response that represents a periodic signal. Since the Laplace transform is a general form of the Fourier transform, the theory they applied can be said to cover a broader theoretical scope than the neural operator centered on FNO, which applies the conventional Fourier transform. Therefore, unlike existing models (Li et al., 2020a; Fanaskov and Oseledets, 2023; Bonev et al., 2023), it can learn damping effects through transient responses.

Since the steady-state response corresponds to a periodic signal in linear or weakly nonlinear systems, the Fourier integral operator can be applied (Li et al., 2020a). We adopt the FNO framework (Li et al., 2020a), which operates in the frequency domain by decomposing input signals into frequency components and applying a learned operator to map them to outputs. As LNO's steady-state response is a superposition of discrete frequency components, it aligns naturally with FNO's mechanism, enhancing expressivity while preserving mathematical consistency. This integration improves efficiency by approximating infinite-frequency expansions with finite computations in the frequency domain while maintaining accuracy. It also captures complex signal dynamics through nonlinear operators, ensuring robust performance across varying frequency resolutions.

To formalize this approach, we start by expressing the LNO solution in its standard decomposition, separating the transient and steady-state components, as shown in Appendix A. This formulation combines the decomposition of LNO with FNO's frequency domain learning capabilities; the transient response, expressed as a sum of decaying exponentials, remains in the time domain, while the steady-state response, originally a sum of discrete frequency components, is processed by a learnable Fourier integral operator $R(\omega_l; \theta)$ (Li et al., 2020a) and mapped back to the time domain, resulting in

$$u(t) = \sum_{n=1}^{N} \gamma_n \exp(\mu_n t) + \sum_{l=-\infty}^{\infty} \lambda_l R(\omega_l; \theta) e^{i\omega_l t}, \tag{1}$$

where all notations are taken from (Cao et al., 2024).

The overall architecture and composition of the layers are shown in Figure 1, in which we follow the notations of (Cao et al., 2024). Our model consists of four transient layers and two steady layers to achieve balanced learning between transient and steady-state responses. The detailed process of Figure 1 is that starting with an input function $\mathbf{f}(t)$, we first lift it to a higher-dimensional representation using a neural network $\mathcal{P}$. Next, we apply the Laplace transform to the resulting higher-dimensional function $v(t)$. We then decompose the transformed output into its transient and steady-state components for each dimension. In the steady layer, we apply the linear transform $R$ on the lower Fourier

modes and filter out the higher modes. Then compute the steady-state response residues $\lambda_l$ based on input poles $i\omega_l$ and residues $\alpha_l$. The local linear transformation $W$ is applied after all the Transient and Steady layers. Finally, we project the output $u(t)$ back to the target dimension using another neural network $\mathcal{Q}$.

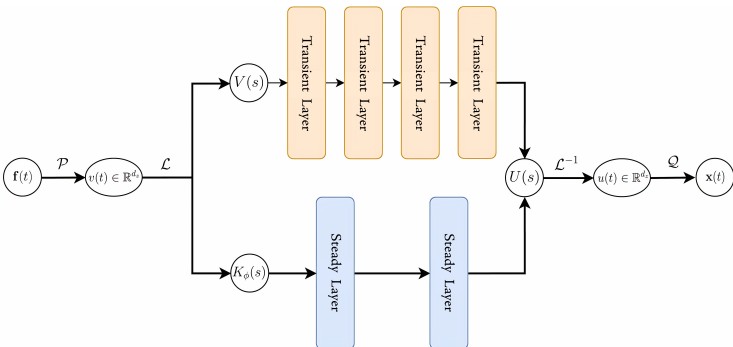

(a) Full Architecture of LFNO

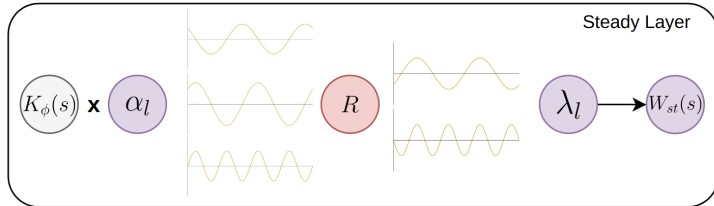

(b) Description of Applying the Fourier integral operator in Steady Layer

Figure 1: Model architecture of LFNO. (a) Overall process: given input function $\mathbf{f}(t)$, 1. apply Laplace transform, 2. take the Transient and the Steady layers for each component, and 3. project the output, $u(t)$. (b) Steady layer structure: apply the pole-residue method, linear transform $R$ on the lower Fourier modes, and filter out the higher modes.

## 3 DATASET GENERATION

We generate our datasets using multiple ODEs and PDEs to measure the performance of LFNO. We adopt three ODEs (Duffing, Lorenz, Pendulum) and five PDEs (Euler-Bernoulli beam, heat, reaction-diffusion, Brusselator, Gray-Scott) for evaluation. The visualization of a sample in each dataset is shown in Appendix D.

### 3.1 ODEs

We generate 380 samples for each ODE dataset by varying the parameter $A$ in the source term $f(t) = Ae^{-0.05t}\sin(5t)$. The values of $A$ are chosen exclusively for each sample from the array of 380 evenly spaced values over $[0.05, 10]$. Each sample has 2048 timesteps with the time interval $\Delta t = 0.01$ seconds. We employ the `solve_ivp` ODE solver in SciPy, using the `RK45` algorithm with a substep of 4.

**Duffing equation**  The Duffing equation is a nonlinear second-order differential equation describing the driven oscillator. The Duffing equation is written as follows.

$$\ddot{x} + c\dot{x} + \alpha x + \beta x^3 = f(t), \tag{2}$$

where $c$ is the damping coefficient, $\alpha$ and $\beta$ are the stiffness of the system, and $x, \dot{x}$, and $\ddot{x}$ are the displacement, velocity, and acceleration of the dynamic response. We set the parameters as $\alpha = 1$ and $\beta = 1$. The initial condition is set to $x(0) = 0$ and $\dot{x}(0) = 0$. We conduct two experiments with the damping coefficient $c$ of 0 and 0.5.

**Pendulum equation** The Pendulum equation models the angle $x(t)$ from the rest position to the pendulum driven by the external force $f(t)$. The equation can be expressed as follows.

$$\ddot{x} + c\dot{x} + \frac{g}{l}\sin(x) = f(t). \tag{3}$$

Here, $g$ is the acceleration due to gravity, $l$ is the length of the pendulum, and $c$ is the damping coefficient. For brevity, we use $g/l = 1$ and the initial condition of $x(0) = 0$ and $\dot{x}(0) = 0$. We experiment with the damping coefficient $c = 0.5$.

**Forced Lorenz system** The forced Lorenz system is the system of three variables $x(t)$, $y(t)$ and $z(t)$ given as

$$\begin{aligned}
\dot{x} &= \gamma(y - x) \\
\dot{y} &= x(\rho - z) - y \\
\dot{z} &= xy - \beta z - f(t),
\end{aligned} \tag{4}$$

where $\gamma$, $\rho$, and $\beta$ are the parameters determining the physical properties of the atmospheric convection, which is what the Lorenz system represents. We train the model with $\rho$ of 5 and 10, while fixing the other parameters as $\gamma = 10$ and $\beta = 8/3$. Both experiments use the same initial conditions $x(0) = 1, y(0) = 0, z(0) = 0$.

### 3.2 PDEs

We generate the datasets using the same method in the ODE experiments. Instead, we apply different forcing functions for each PDE experiment. We solve the PDEs using the same solver settings as in the ODE experiments, with one exception: the reaction-diffusion equation, which employs the `BDF` algorithm. Throughout the PDE experiments, the resolution of the spatio-temporal grid is fixed as $50 \times 50$. The details of the PDE configurations are described in Table 1. For the PDEs, all boundary conditions are Dirichlet except for the Brusselator equation, which uses a Neumann boundary condition.

**Euler-Bernoulli beam equation** The deflection $y(x, t)$ of a 1-dimensional homogeneous Euler-Bernoulli beam with the forcing function $f(x, t)$ is given as follows:

$$EI\frac{\partial^4 y}{\partial x^4} + \rho A\frac{\partial^2 y}{\partial t^2} = f(x, t), \tag{5}$$

where $EI$ is the bending stiffness, $\rho$ is the density of the beam, and $A$ is a cross-sectional area of the beam. The forcing function is defined as $f(x, t) = -99Ae^{-x}\sin(10t)$, and the parameters of $EI = 1.334$, $\rho = 7850$, and $A = 0.01$ are used.

**Heat equation** The heat equation describes how a heat quantity $y(x, t)$ in a given region changes over time through diffusion, which is equivalent to the diffusion equation of zero bulk velocity. The equation is given as follows:

$$\frac{\partial^2 y}{\partial x^2} - \frac{\partial y}{\partial t} = f(x, t). \tag{6}$$

Here, we use the source term of $f(x, t) = Ae^{-t}(1 - \pi^2)\sin(\pi x)$.

**Reaction-diffusion equation** The reaction-diffusion equations are utilized to study the concentration of chemicals under reaction over time. In this experiment, we used the governing equation as follows:

$$D\frac{\partial^2 y}{\partial x^2} + ky^2 - \frac{\partial y}{\partial t} = f(x, t), \tag{7}$$

where $y(x, t)$ is a concentration of the chemical, $D$ is the diffusion coefficient, and $k$ is the rate of reaction. The source term $f(x, t)$ is defined as $f(x, t) = Ae^{-t}(1 - \pi^2)\sin(\pi x) + A^2 e^{-2t}\sin(\pi x)^2$. The coefficients are set to $D = 1 - 0.95\pi^2$ and $k = 1$.

**Brusselator equation**  The Brusselator equation describes concentrations of two chemicals, denoted as $u$ and $v$, in an autocatalytic reaction. The system is governed by

$$\frac{\partial u}{\partial t} = D_0 \frac{\partial^2 u}{\partial x^2} + a + f(x,t) - (1+b)u + vu^2$$
$$\frac{\partial v}{\partial t} = D_1 \frac{\partial^2 v}{\partial x^2} + bu - vu^2. \tag{8}$$

Here, $D_0$ and $D_1$ are diffusibility parameters, and $a$ and $b$ are the parameters related to the concentration of chemicals. We use the same source term of $f(x,t) = Ae^{-t}(1 - \pi^2)\sin(\pi x)$ and set the parameters as $D_0 = 1$, $D_1 = 0.5$, $a = 1$, and $b = 3$.

**Gray-Scott equation**  The Gray-Scott equation represents the concentration of two chemicals $u$ and $v$, in a specific reaction involving three chemicals. We slightly alter the original PDE to reduce the dimensionality of the domain and introduce the source term $f(x,t)$, which is the same function used in the reaction-diffusion experiment. The governing equation is given as

$$\frac{\partial u}{\partial t} = r_u \frac{\partial^2 u}{\partial x^2} - uv^2 + F(1-u) + \frac{1}{10}f(x,t)$$
$$\frac{\partial v}{\partial t} = r_v \frac{\partial^2 v}{\partial x^2} + uv^2 - (F+K)v + \frac{1}{10}f(x,t), \tag{9}$$

where $r_u$ and $r_v$ are the diffusion coefficients; $F$ and $K$ are the coefficients that determine the characteristic of the reaction. We set $r_u = 0.01$, $r_v = 0.02$, $F = 0.04$, and $K = 0.06$.

| Task | Boundary conditions | Initial values (for all $x$) | $\Delta x$ | $\Delta t$ |
|---|---|---|---|---|
| Euler-Bernoulli beam | $y = 0$ for each end | $y(x,0) = 0$ $\dot{y}(x,0) = 0$ | 0.03 | 0.02 |
| Heat equation | | $y(x,0) = 0$ | 0.08 | 0.01 |
| Reaction-diffusion | | | 0.08 | 0.02 |
| Brusselator | Neumann | $u(x,0) = 0.5$ $v(x,0) = 1$ | 0.03 | 0.02 |
| Gray-Scott | Dirichlet ends | $u(x,0) \sim \mathcal{U}(0,1)$ $v(x,0) \sim \mathcal{U}(0,1)$ | 0.03 | 0.02 |

Table 1: Detailed configurations of PDEs

## 4  RESULTS

The theoretical process of LFNO is well translated into empirical effectiveness. In this section, we present the supporting empirical results by comparing LNO, LFNO, and FNO side by side. Other specialized neural operators are excluded, as we aim to empirically compare the three related general-purpose neural operators in various problem settings. We calculate the relative error $\mathcal{L}_2$ for the prediction of the test samples and use it as a quantitative metric. We then compare the model output of LFNO and others against the same sample, using them as a qualitative metric. We conduct experiments with a fixed number of epochs for each task. We present the experimental results below and provide implementation details in Appendix B.

### 4.1  ODE EXPERIMENTS

The detailed $\mathcal{L}_2$ errors are described in the left panel of Table 3. This indicates that LFNO shows a significant difference from FNO in ODE experiments. LFNO also exhibits superior performance compared to LNO. For qualitative evaluation, we overlay the inference results of the baseline models on those of LFNO, as shown in Table 2. In the Lorenz experiments, LFNO captures the damping effect with high fidelity, whereas both LNO and FNO fail to reproduce it. Furthermore, LFNO demonstrates a more effective ability to capture periodic features compared to LNO in both the Duffing $c = 0.5$ and Pendulum $c = 0.5$ experiments.

| Task | LFNO vs. FNO | LFNO vs. LNO |
|------|--------------|--------------|
| Duffing $c = 0$ | | |
| Duffing $c = 0.5$ | | |
| Lorenz $\rho = 5$ | | |
| Lorenz $\rho = 10$ | | |
| Pendulum $c = 0.5$ | | |

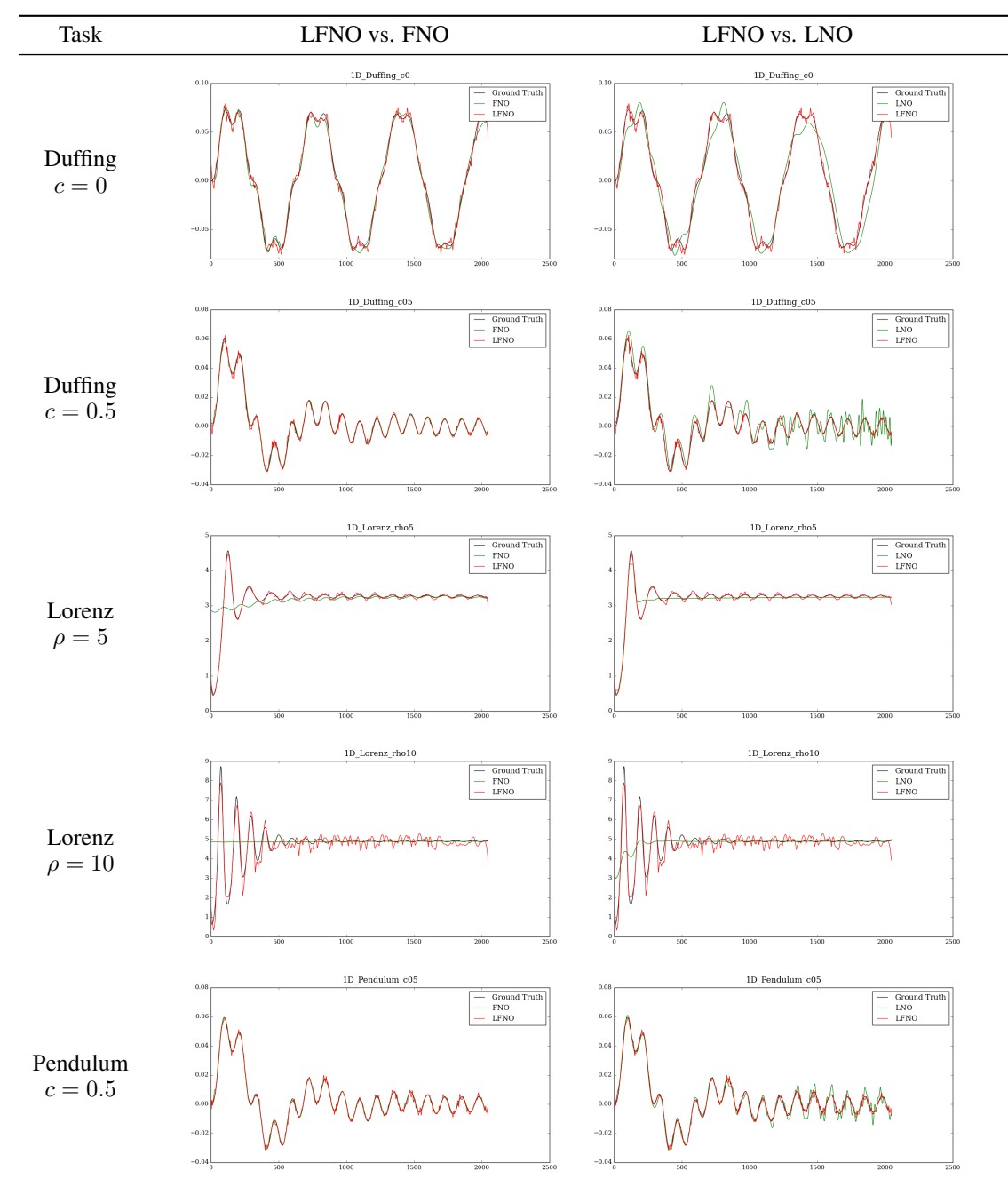

Table 2: Qualitative results for three models. Comparison of LFNO with FNO (left) and LNO (right). Red: LFNO results; Green: comparison model results.

## 4.2 PDE EXPERIMENTS

The detailed $\mathcal{L}_2$ errors are described in the right panel of Table 3. The results show that our model achieves performance comparable to FNO, while exhibiting considerable differences from LNO. The complete qualitative comparison plots are presented in Figure 2, with full-size figures available in Appendix E. Overall, the results demonstrate that our model outperforms LNO qualitatively and illustrates performance comparable to FNO. In the right panel of Figure 2, we present a scatter plot of $y = x$ to illustrate how closely the predictions match the ground truth, where points closer to

| Equation | ODEs | | | | | PDEs | | | | |
|---|---|---|---|---|---|---|---|---|---|---|
| | Duffing $c = 0$ | Duffing $c = 0.5$ | Lorenz $\rho = 5$ | Lorenz $\rho = 10$ | Pendulum $c = 0.5$ | Beam | Heat | Reaction Diffusion | Brusselator | Gray-scott |
| Epochs | 6200 | 5200 | 8100 | 4000 | 6800 | 4900 | 4700 | 4700 | 5000 | 4800 |
| FNO | 0.9998 | 0.0565 | 0.1225 | 0.4340 | 0.0426 | **0.0035** | **0.0017** | **0.0030** | **0.0024** | **0.0138** |
| LNO | 0.2014 | 0.0762 | 0.0288 | 0.2571 | 0.0735 | 0.2509 | 0.0227 | 0.0779 | 0.0137 | 0.1709 |
| LFNO | **0.1288** | **0.0314** | **0.0075** | **0.1731** | **0.0315** | 0.0153 | 0.0051 | 0.0049 | 0.0041 | 0.0155 |

Table 3: $\mathcal{L}_2$ error table for ODEs and PDEs tasks.

the line $y = x$ indicate higher precision. The scatter plot shows that LFNO and FNO produce predictions closely aligned with the $y = x$ line, indicating comparable accuracy, whereas LNO displays a more dispersed distribution, reflecting lower performance. In particular, see Figure 2c, we confirm that LNO breaks down completely, failing to produce any meaningful predictions, evident in the difference between the two plots on the left and the non-diagonal scatter plot on the right.

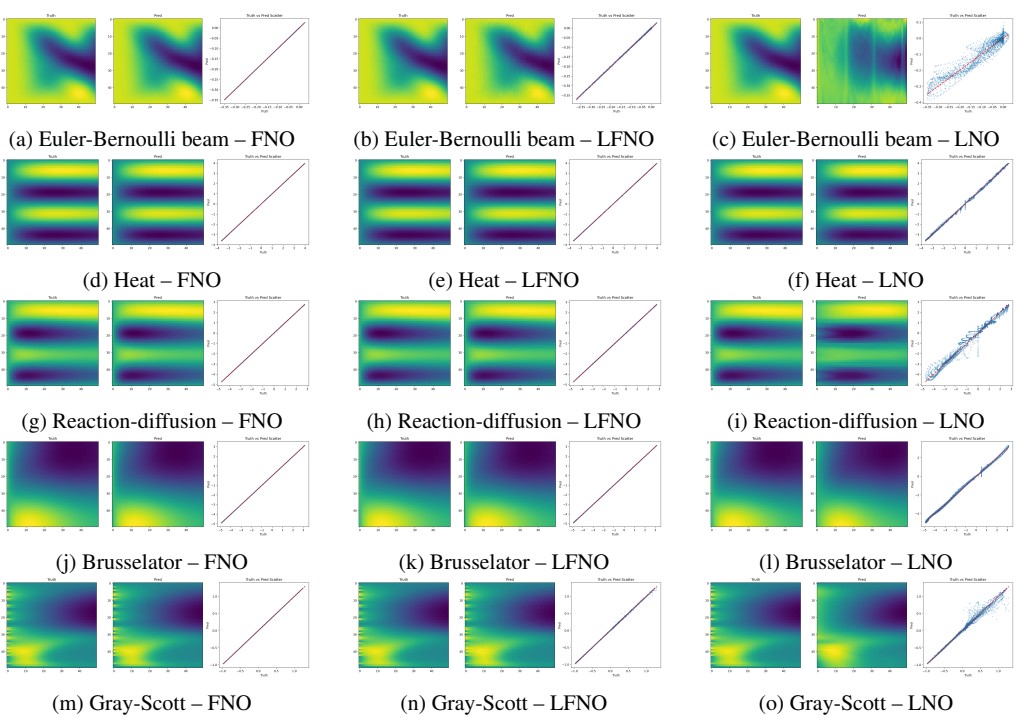

(a) Euler-Bernoulli beam – FNO  (b) Euler-Bernoulli beam – LFNO  (c) Euler-Bernoulli beam – LNO

(d) Heat – FNO  (e) Heat – LFNO  (f) Heat – LNO

(g) Reaction-diffusion – FNO  (h) Reaction-diffusion – LFNO  (i) Reaction-diffusion – LNO

(j) Brusselator – FNO  (k) Brusselator – LFNO  (l) Brusselator – LNO

(m) Gray-Scott – FNO  (n) Gray-Scott – LFNO  (o) Gray-Scott – LNO

Figure 2: Qualitative results and scatter plots for five PDEs (Euler-Bernoulli beam, Heat, Reaction–Diffusion, Brusselator, Gray-Scott) using FNO, LFNO, and LNO. Rows correspond to PDEs, columns correspond to the three models. The right panel of each subfigure shows the (truth, prediction) scatter, where points aligned closer to $y = x$ indicate overall higher accuracy.

### 4.3 LEARNING CURVES

We compare the training dynamics of FNO, LNO, and LFNO by analyzing the learning curves. The overall learning curves are shown in Figure 3 and Figure 4. Across all experiments, training, and validation sets are represented by lime and red for LFNO, orange and blue for LNO, and magenta and cyan for FNO.

In ODE experiments, we observe that FNO training fails in certain cases, most notably in Figure 3a, Figure 3c, and Figure 3d. We also confirm that LNO has unstable training in Figure 3c and Figure 3d. In general, our model demonstrates more stable and reliable learning behavior compared to both LNO and FNO in ODE tasks.

For the PDE experiments, the training and validation losses of three models illustrate notable differences. In particular, LNO exhibits substantially higher losses than LFNO and FNO, suggesting training difficulties in Figure 4a and Figure 4c. Throughout these experiments, our model and FNO consistently exhibit stable convergence, whereas LNO struggles to achieve comparable results.

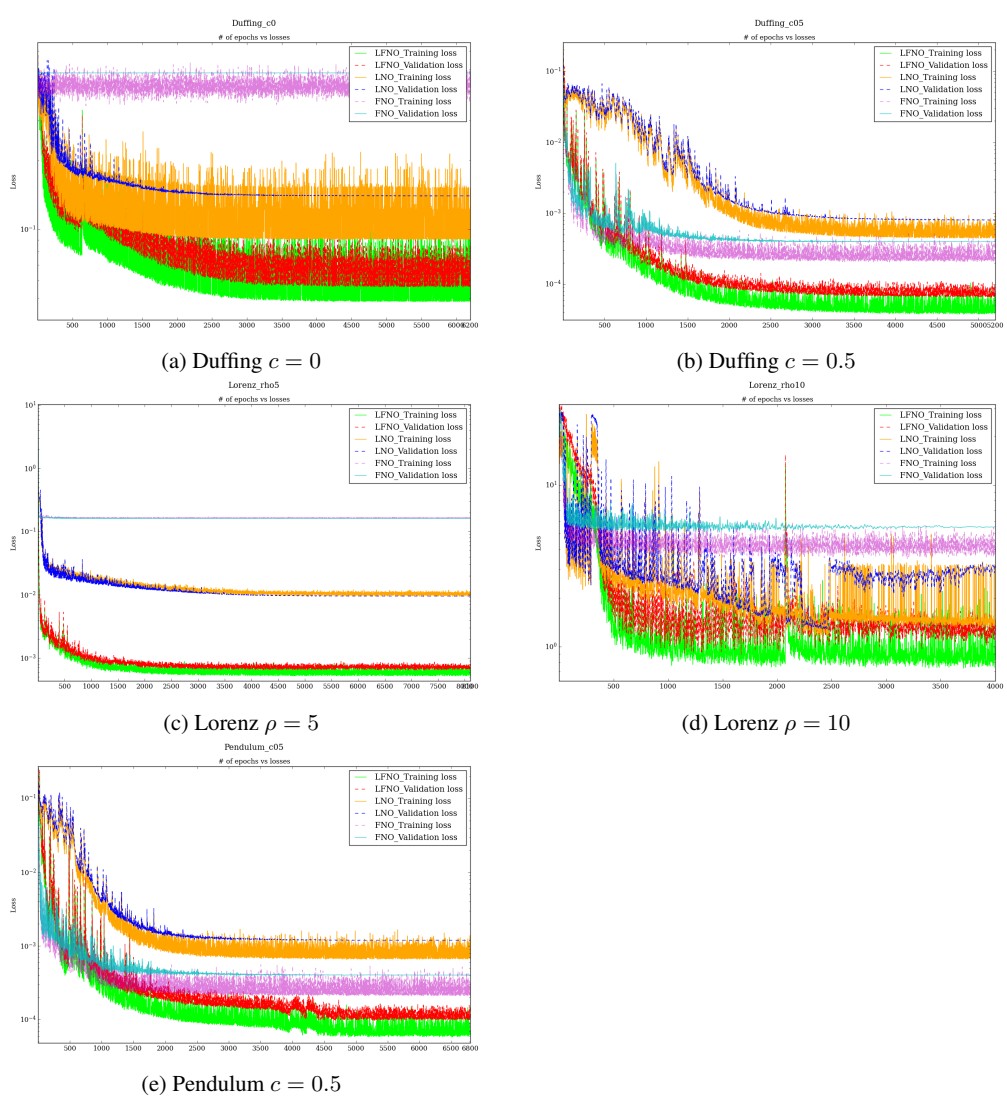

(a) Duffing $c = 0$

(b) Duffing $c = 0.5$

(c) Lorenz $\rho = 5$

(d) Lorenz $\rho = 10$

(e) Pendulum $c = 0.5$

Figure 3: ODE Learning curves on different models. Training and validation sets are represented by lime and red for LFNO, orange and blue for LNO, and magenta and cyan for FNO.

## 5 DISCUSSION

(Cao et al., 2024) claimed that LNO produces lower validation errors than FNO, but our actual experimental results showed otherwise. Interestingly, despite its stated ability to capture transient behavior, LNO exhibited a larger performance gap relative to our model in the beam equation experiments compared to the ODE experiments. Theoretically, it was well approached through the Laplace transform and pole-residue formulation, but it appeared to have encountered limitations during the engineering phase.

The original LNO datasets used different functions for the train and validation sets. This resulted in a change to the target mapping during model validation. Thus, we constructed datasets in which the functions of the training and validation sets were aligned. Among the datasets we generate, the

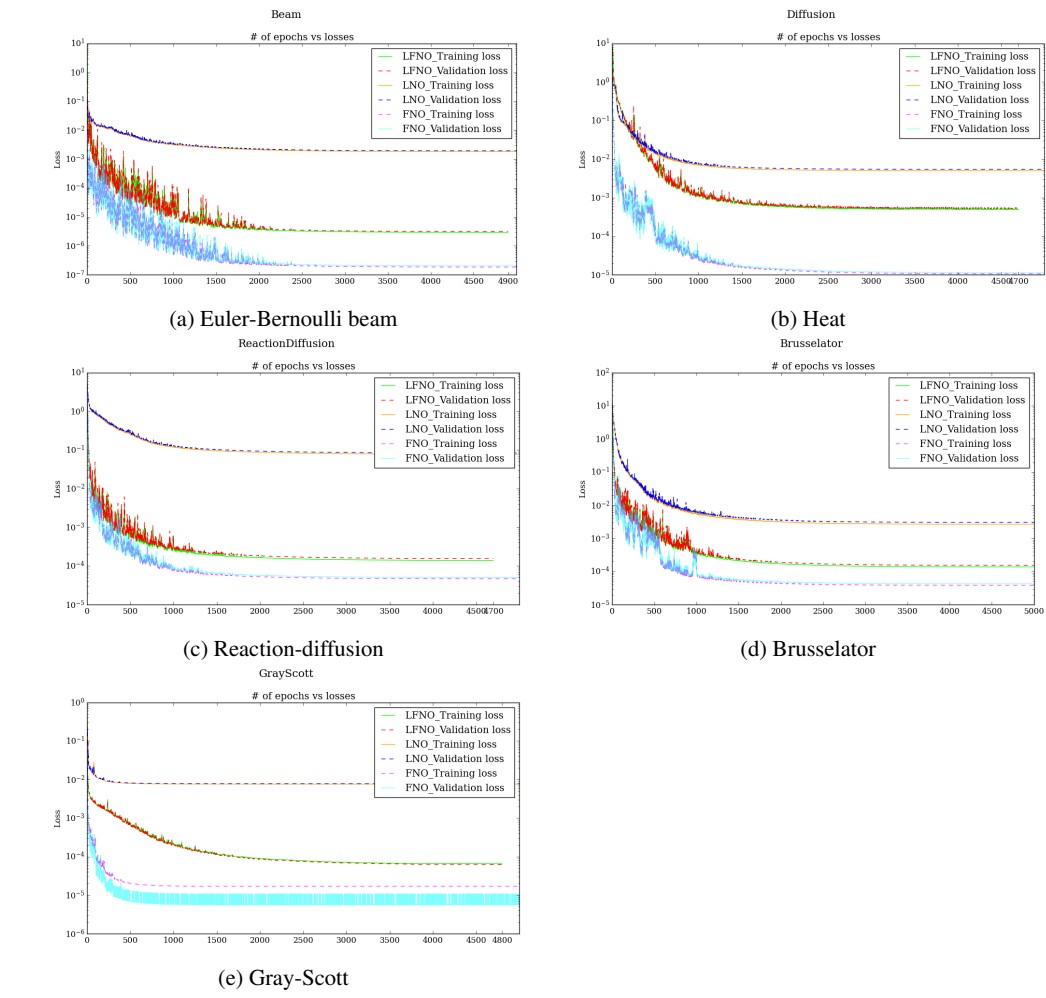

Figure 4: PDE Learning curves on different models. Training and validation sets are represented by lime and red for LFNO, orange and blue for LNO, and magenta and cyan for FNO.

PDE beam equation exhibits substantially stronger transient effects than those found in conventional LNO datasets. We suspect that the data used for comparative experiments may not fully capture the transient behavior.

All ODEs examined in this study indicate that the steady-state response becomes chaotic under certain conditions. Future work, focusing on developing methodologies to approximate chaotic equations and to embed chaotic regions within LFNO, may also illustrate the novel utility of bridging Laplace and Fourier for neural operator learning.

## 6 CONCLUSION

In this work, we propose LFNO, a unified framework that integrates LNO and FNO. Within this formulation, the transient response is captured by the LNO, while the Fourier integral operator enhances the steady-state response. Introducing nonlinearity into the steady-state component enables a broader representation of signals. Moreover, our model can be further enhanced to capture transient characteristics, allowing it to learn such behaviors more effectively than LNO. Consequently, our results demonstrate that LFNO not only achieves a theoretical integration of LNO and FNO but also provides empirically robust learning performance. These results demonstrate that LFNO offers clear advantages as an operator learning framework for linear and nonlinear DEs with transient states, while showing performance comparable to that of existing models.

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

## A  MATHEMATICAL FORMULATION

All notations are taken from (Cao et al., 2024).

$$u(t) = \sum_{n=1}^{N} \gamma_n e^{\mu_n t} + \sum_{l=-L}^{L} \lambda_l e^{i\omega_l t}, \tag{10}$$

where the first summation term is the transient response and the second summation term is the familiar steady-state response. Hence, the above equation could be simplified as $u(t) = u_{transient}(t) + u_{steady}(t)$. Since the steady-state response is represented in the frequency domain, we can apply the Fast Fourier transform to the steady-state response. Applying the Fourier transform to $u_{steady}(t)$ yields the following result.

$$\mathcal{F}(u_{steady})(\omega) = \int_{-\infty}^{\infty} \left( \sum_{l=-\infty}^{\infty} \lambda_l e^{i\omega_l t} \right) e^{-i\omega t} dt. \tag{11}$$

Assuming that the function $u_{steady}(t)$ satisfies the Dirichlet condition, we can change the order of summation and integration. Therefore, we formulate $u_{steady}(t)$ as follows:

$$\mathcal{F}(u_{steady})(\omega) = \int_{-\infty}^{\infty} \left( \sum_{l=-\infty}^{\infty} \lambda_l e^{i\omega_l t} \right) e^{-i\omega t} dt = \sum_{l=-\infty}^{\infty} \lambda_l \int_{-\infty}^{\infty} e^{i\omega_l t} e^{-i\omega t} dt \tag{12}$$

The integral in Equation (12) is formally divergent as a classical integral. Therefore, it is understood in the sense of distributions, yielding the Dirac delta function:

$$\int_{-\infty}^{\infty} e^{i\omega_l t} e^{-i\omega t} dt = \int_{-\infty}^{\infty} e^{i(\omega_l - \omega)t} dt = 2\pi \, \delta(\omega_l - \omega), \tag{13}$$

where the factor $2\pi$ appears according to the Fourier transform convention used. Using the sifting property of the Dirac delta, we can then write

$$\mathcal{F}(u_{steady})(\omega) = \sum_{l=-\infty}^{\infty} \lambda_l \cdot \delta(\omega_l - \omega), \tag{14}$$

which explicitly represents the discrete frequency components of the steady-state response. We now apply the Fourier integral operator (Li et al., 2020a) to these frequency components. FNO learns a non-linear operator $R(\omega; \theta)$ in the frequency domain, where $\theta$ denotes learnable parameters, mapping each input frequency component to the corresponding output. The output frequency components are

$$
\begin{aligned}
\hat{u}_{out}(\omega) &= R(\omega; \theta) \cdot \mathcal{F}(u_{steady})(\omega) \\
&= R(\omega; \theta) \cdot \sum_{l=-\infty}^{\infty} \lambda_l \cdot \delta(\omega_l - \omega) \\
&= \sum_{l=-\infty}^{\infty} \lambda_l \cdot R(\omega_l; \theta) \cdot \delta(\omega_l - \omega),
\end{aligned}
\tag{15}
$$

where we have applied the sifting property of the Dirac delta to evaluate $R(\omega; \theta)$ at $\omega = \omega_l$. Finally, applying the inverse Fourier transform yields the time-domain output:

$$
u_{out}(t) = \mathcal{F}^{-1}(\hat{u}_{out})(\omega) = \sum_{l=-\infty}^{\infty} \lambda_l R(\omega_l; \theta) e^{i\omega_l t}.
\tag{16}
$$

Thus, the full solution combining the transient and steady-state responses can be written as

$$
u(t) = \sum_{n=1}^{N} \gamma_n e^{\mu_n t} + \sum_{l=-L}^{L} \lambda_l R(\omega_l; \theta) e^{i\omega_l t}.
\tag{17}
$$

## B  IMPLEMENTATION DETAILS

In this section, we introduce the implementation details. For all models, the Adam optimizer with step learning rate scheduling is used to improve stability and better convergence. We trained the model on an NVIDIA RTX 3090 GPU. The model hyperparameters are listed in Table 4. We particularly employed the early stopping method for all experiments and compared the corresponding LNOs with those from the same number of epochs iterated in the LNO experiment. We also present the computational cost for each training in Table 5.

| Application | FNO ODEs | LNO ODEs | LFNO ODEs | FNO PDEs | LNO PDEs | LFNO PDEs |
|---|---|---|---|---|---|---|
| Layer | 4 | 1 | 6 | 4 | 1 | 6 |
| Width | 4 | 4 | 4 | 16 | 16 | 16 |
| Mode | 16 | 16 | 16 | 4 | 4 | 4 |
| Train batch size | 64 | 64 | 64 | 50 | 50 | 50 |
| Learning rate | 0.0025 | 0.0025 | 0.0025 | 0.002 | 0.002 | 0.002 |
| weight decay | 0.01 | 0.01 | 0.01 | 0.0001 | 0.0001 | 0.0001 |
| scheduler step size | 100 | 100 | 100 | 100 | 100 | 100 |
| gamma | 0.85 | 0.85 | 0.85 | 0.5 | 0.5 | 0.8 |
| Activation function | relu | sin | relu | relu | sin | relu |

Table 4: Hyperparameters used in the FNO, LNO, and LFNO for training.

## C  ABLATION STUDY

We conduct the following experiment to investigate how the prediction error varies with respect to the number of transient layers and steady-state layers in Table 6.

## D  DATASET SAMPLES

Table 7 shows the dataset visualizations for each task. The datasets themselves have no split, as we used K-fold or a random split on the fly to shuffle and split the test and validation samples.

| Computational cost | FNO | LNO | LFNO |
|---|---|---|---|
| Parameter count | 7,537 | 1,309 | 3,417 |
| Inference time (ms) | 1.767 | 1.145 | 4.150 |
| Multiply-Acculumlate | 146.95 | 2.80 | 16.48 |
| Memory usage | 286.26 | 147.85 | 329.25 |

Table 5: Computational cost for FNO, LNO and LFNO training.

| Type | Duffing_c0 | Duffing_c05 | Lorenz_rho5 | Lorenz_rho10 | Pendulum_c05 |
|---|---|---|---|---|---|
| FNO | 0.2828 | 0.0931 | 0.0196 | 0.2587 | 0.1042 |
| LNO | 0.2014 | 0.0762 | 0.0288 | 0.2571 | 0.0735 |
| LFNO (T1S4) | 0.1508 | 0.0604 | 0.0038 | 0.0990 | 0.0691 |
| LFNO (T2S4) | 0.1597 | 0.0609 | 0.0050 | 0.0747 | 0.0763 |
| LFNO (T2S2) | 0.3173 | 0.0949 | 0.0066 | 0.1293 | 0.0973 |
| LFNO (T4S1) | 0.7744 | 0.4088 | 0.0110 | 0.7586 | 0.4115 |
| LFNO (T4S2) | 0.1288 | 0.0314 | 0.0074 | 0.1731 | 0.0315 |
| LFNO (T4S4) | 0.1758 | 0.0688 | 0.0081 | 0.0928 | 0.0928 |

Table 6: Prediction error comparison across model layers.

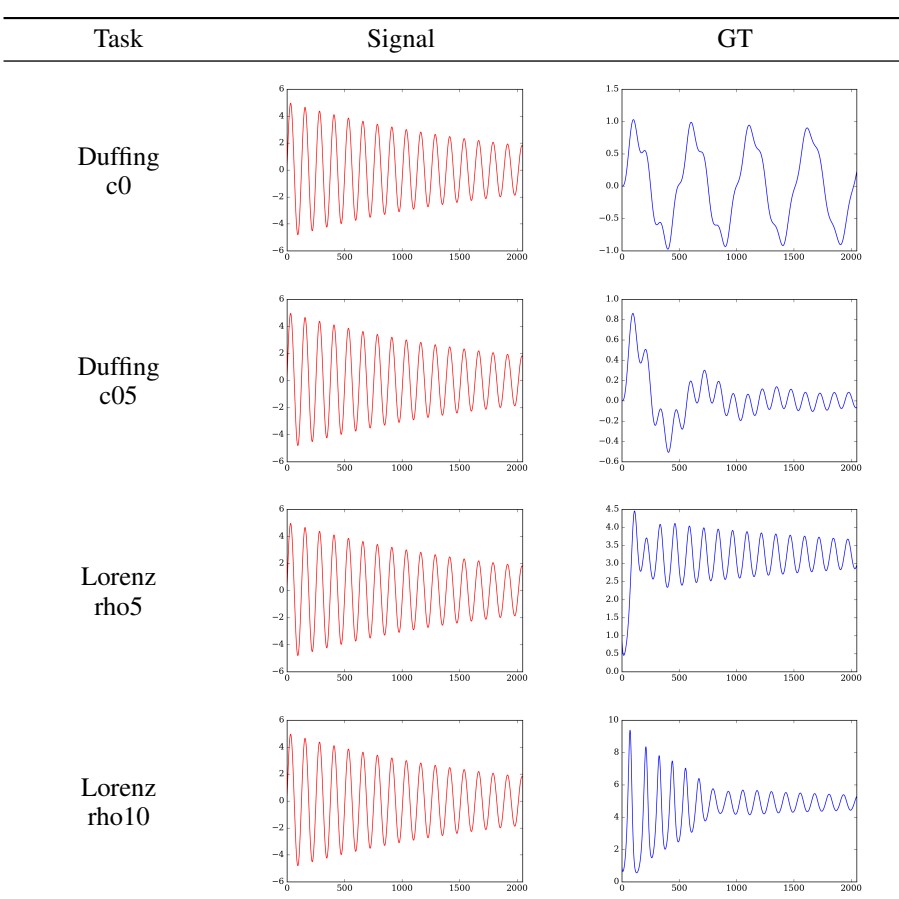

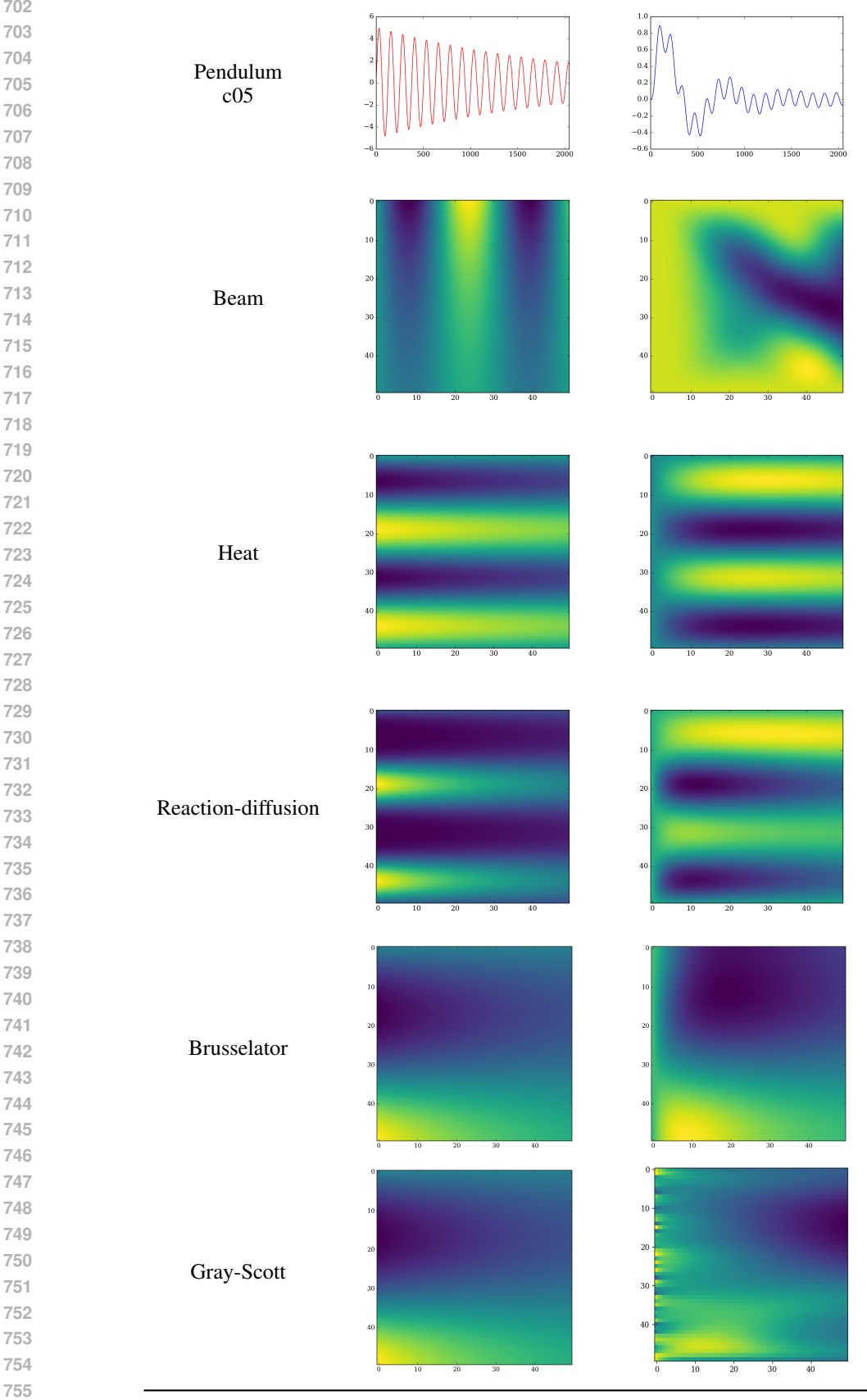

Table 7: Visualization of samples in our dataset

# E   PDEs QUALITATIVE FIGURES

The qualitative results of PDEs are shown in Figure 5, Figure 6, Figure 7, Figure 8, and Figure 9. It can be seen that our model results are as good as the results of FNO in the qualitative representation.

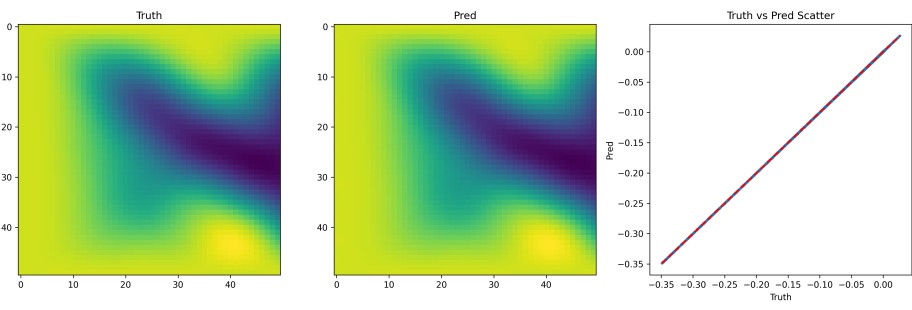

(a) Qualitative Results and Scatter of the Euler-Bernoulli beam in FNO

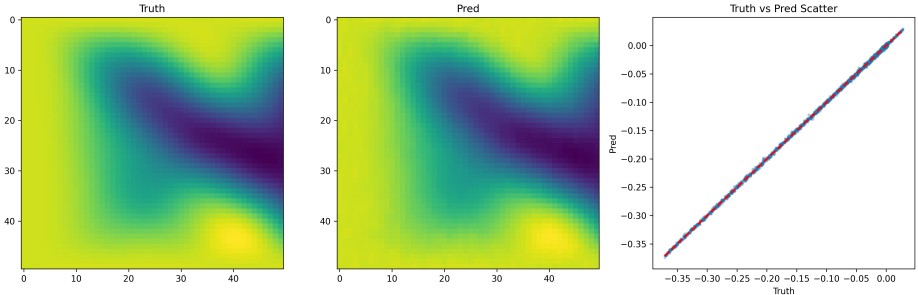

(b) Qualitative Results and Scatter of the Euler-Bernoulli beam in LFNO

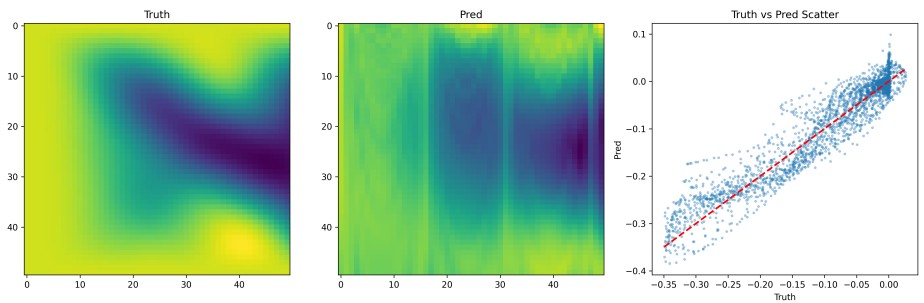

(c) Qualitative Results and Scatter of the Euler-Bernoulli beam in LNO

Figure 5: Euler-Bernoulli beam equation: Comparison of qualitative results computed by FNO, LFNO, and LNO. It represents the ground truth shown in the left plot, the prediction of each model in the center plot, and the (truth, pred) scatters are shown in the right plot.

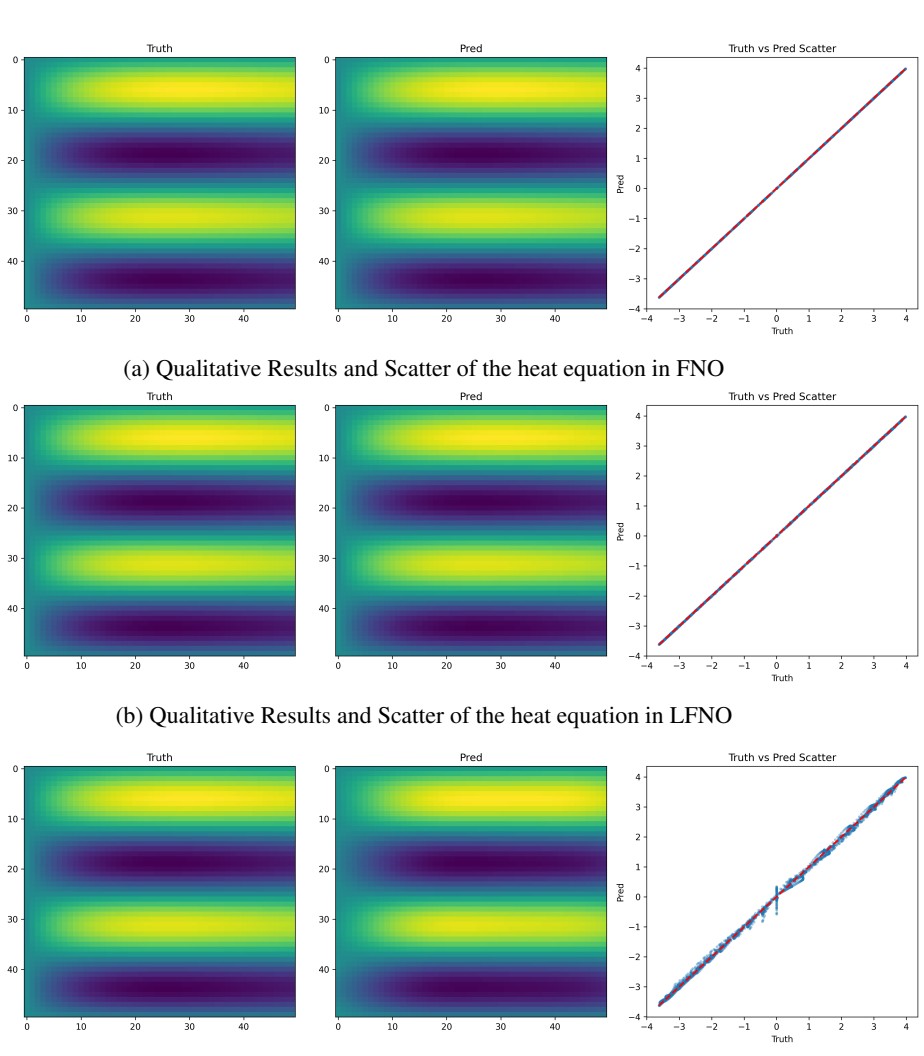

(a) Qualitative Results and Scatter of the heat equation in FNO

(b) Qualitative Results and Scatter of the heat equation in LFNO

(c) Qualitative Results and Scatter of the heat equation in LNO

Figure 6: Heat equation: Comparison of qualitative results computed by FNO, LFNO, and LNO. It represents the ground truth shown in the left plot, the prediction of each model in the center plot, and the (truth, pred) scatters are shown in the right plot.

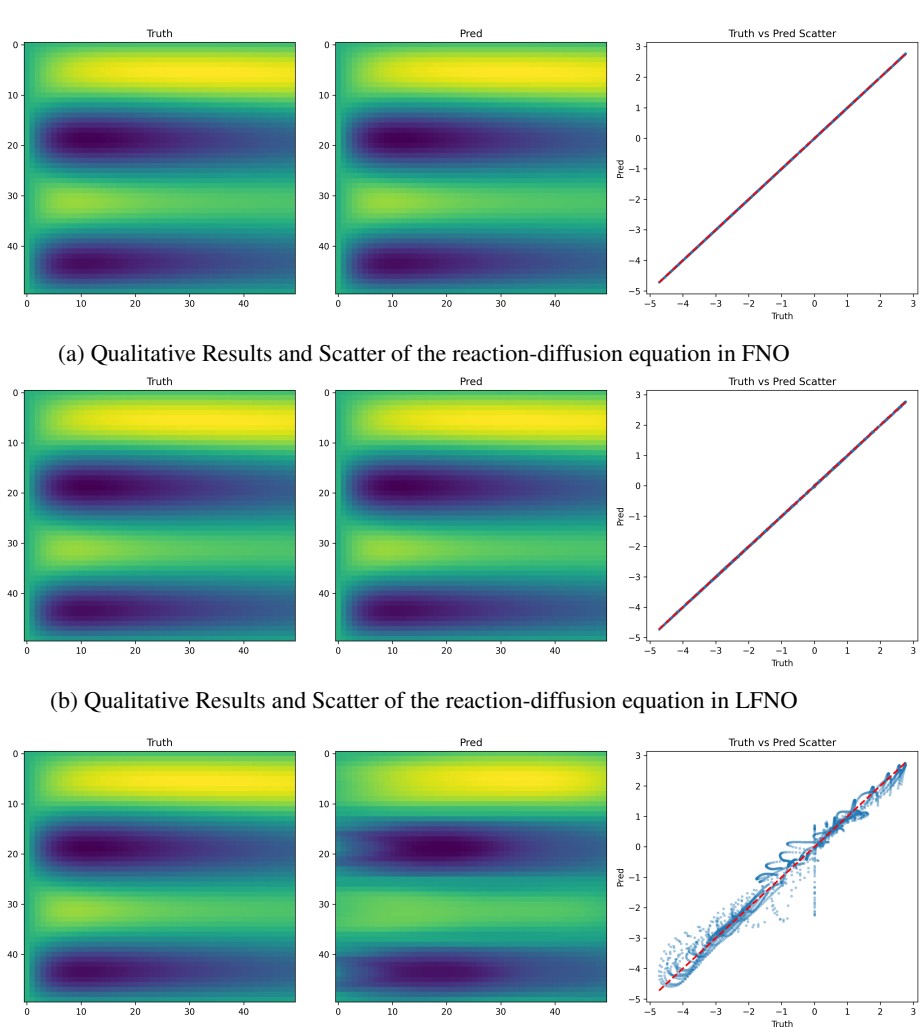

(a) Qualitative Results and Scatter of the reaction-diffusion equation in FNO

(b) Qualitative Results and Scatter of the reaction-diffusion equation in LFNO

(c) Qualitative Results and Scatter of the reaction-diffusion equation in LNO

Figure 7: Reaction-diffusion equation: Comparison of qualitative results computed by FNO, LFNO, and LNO. It represents the ground truth shown in the left plot, the prediction of each model in the center plot, and the (truth, pred) scatters are shown in the right plot.

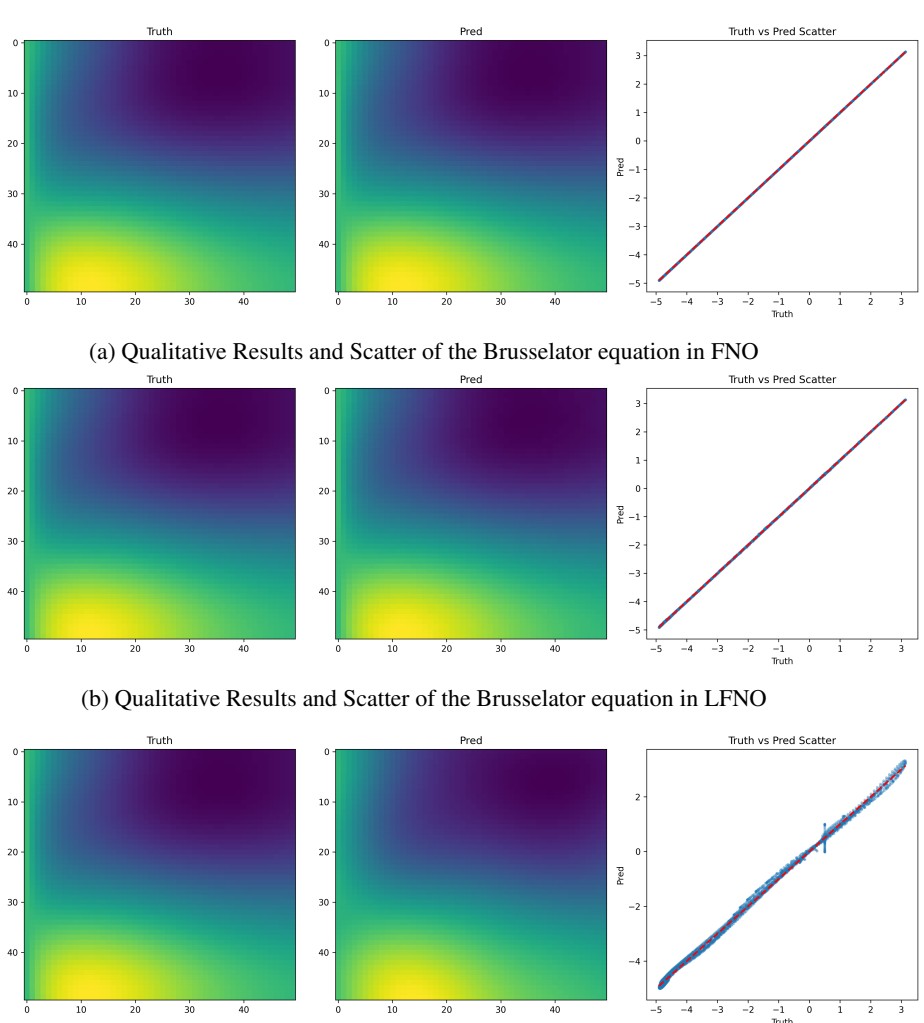

(a) Qualitative Results and Scatter of the Brusselator equation in FNO

(b) Qualitative Results and Scatter of the Brusselator equation in LFNO

(c) Qualitative Results and Scatter of the Brusselator equation in LNO

Figure 8: Brusselator equation: Comparison of qualitative results computed by FNO, LFNO, and LNO. It represents the ground truth shown in the left plot, the prediction of each model in the center plot, and the (truth, pred) scatters are shown in the right plot.

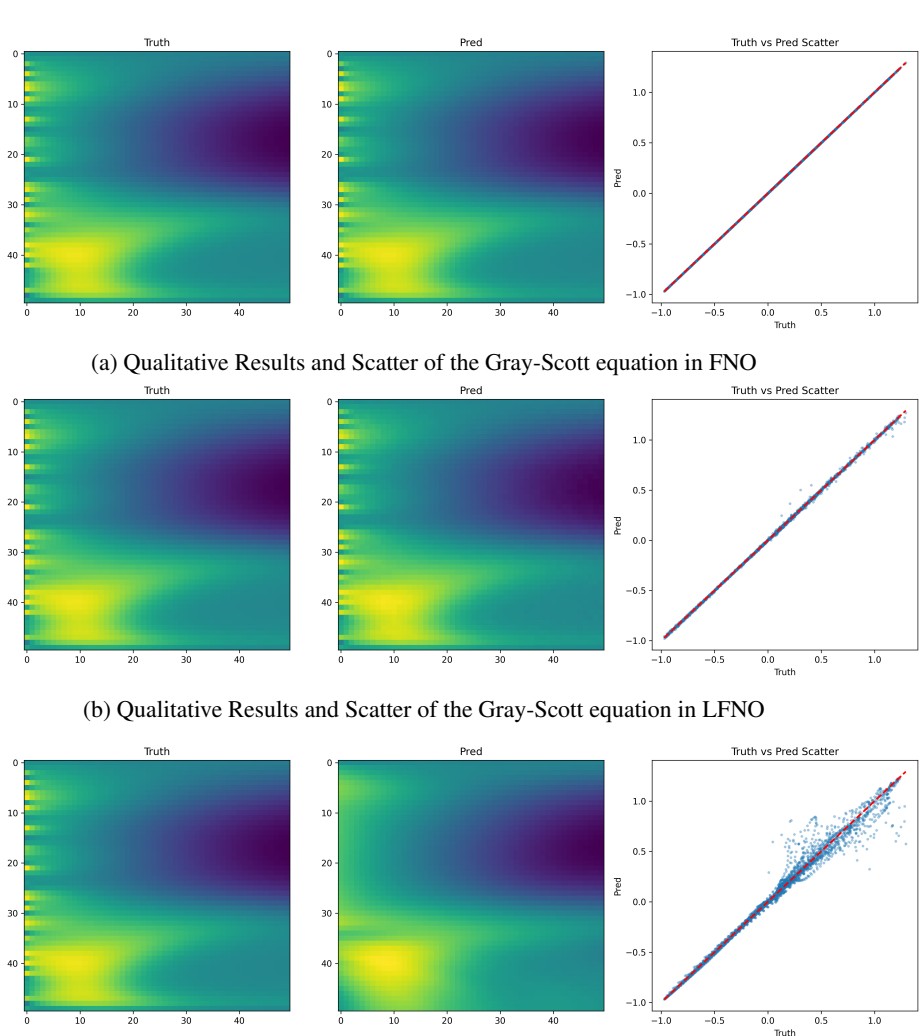

(a) Qualitative Results and Scatter of the Gray-Scott equation in FNO

(b) Qualitative Results and Scatter of the Gray-Scott equation in LFNO

(c) Qualitative Results and Scatter of the Gray-Scott equation in LNO

Figure 9: Gray-Scott equation: Comparison of qualitative results computed by FNO, LFNO, and LNO. It represents the ground truth shown in the left plot, the prediction of each model in the center plot, and the (truth, pred) scatters are shown in the right plot.

