# OpenReview forum: "LFNO: Bridging Laplace and Fourier for Effective Operator Learning"
_ICLR.cc/2026/Conference — Submitted to ICLR 2026_

### Official Review · Reviewer_NutN · 2025-10-26

**Soundness:** 2
**Presentation:** 3
**Contribution:** 2
**Rating:** 4
**Confidence:** 4

**Summary:**

This paper proposes LFNO, a hybrid operator-learning framework that unifies the strengths of the Laplace Neural Operator (LNO) and the Fourier Neural Operator (FNO). The main idea is that LNO captures transient (exponentially decaying) responses, while FNO efficiently models steady-state (periodic) dynamics. LFNO combines them through a dual-branch architecture—one Laplace branch for transient dynamics and one Fourier branch for steady components—whose outputs are fused in the latent space. Experiments on canonical ODE and PDE benchmarks show that LFNO achieves stable performance and moderate improvements over both FNO and LNO.

**Strengths:**

The motivation of bridging Laplace and Fourier domains is clear and theoretically sound, as the two transforms respectively handle transient and steady-state behaviors. The architecture design is reasonable and easy to implement within existing operator-learning frameworks. The results are consistent across multiple benchmarks, showing marginal but steady improvements in accuracy and stability over FNO and LNO, which demonstrates that the hybridization is technically effective.

**Weaknesses:**

(1) Limited conceptual novelty and incremental contribution. The proposed LFNO mainly combines two existing operator-learning paradigms (FNO and LNO) through a dual-branch fusion structure. While the formulation is coherent and empirically justified, it does not introduce a fundamentally new operator formulation, learning principle, or theoretical insight.

(2) Simplified experimental settings and lack of complex operator-learning scenarios. The experiments are conducted primarily on standard ODE/PDE benchmarks, which are relatively simple and well-established in the literature. Many tasks are less challenging than those used in prior FNO works. This limits the demonstration of LFNO’s robustness and practical applicability.

(3) While LFNO achieves lower errors than FNO and LNO, the paper does not provide deeper analysis that reveal why the Laplace–Fourier combination works, or what specific benefits it brings beyond marginal accuracy improvements. As a result, the results alone do not fully substantiate the claimed conceptual innovation.

**Questions:**

(1) How sensitive is the performance to the fusion weighting between the two branches?

(2) Can LFNO handle the same experimental settings used in the original FNO paper? If so, how does it perform compared to FNO in these scenarios?

---

> ### Author Response · Authors · 2025-12-03
> **Response to Reviewer NutN**
>
> We thank reviewer NutN for the detailed review and constructive questions. Our responses are as follows:
>
> 1. $\textbf{Limited Conceptual Novelty and Incremental Contribution}$
>    - Existing neural operators have typically been designed to handle a single type of condition or regime within a given system. In contrast, we propose a hybrid architecture that first decomposes the solution into transient and steady-state components using LNO and subsequently applies a Fourier transform–based operator to the steady-state component. This establishes, for the first time, that a principled combination of Laplace-Fourier-based neural operators is not only feasible but also theoretically well-motivated.
>
> 2. $\textbf{Simplified experimental settings and lack of complex operator-learning scenarios}$
>    -  Our primary focus was to validate the proposed assumptions on PDEs with a clearly identifiable decomposition structure, ensuring that the methodology operates as intended.
>
> 3. $\textbf{Question 1. How sensitive is the performance to the fusion weighting between the two branches?}$
>     - Thank you for highlighting this issue. When the number of transient layers is fixed, increasing the number of steady layers initially shows a sharp difference. Still, this difference becomes negligible when the number of layers increases from 3 to 4. When the number of steady layers is fixed, increasing the number of transient layers does not show a significant difference. It could be confirmed in Table 6 in Appendix C.
>
> 4. $\textbf{Question 2. Can LFNO handle the same experimental settings used in the original FNO paper?}$
>    - We appreciate this question. We only conducted the experiments that were at LNO. The dataset used in FNO studies posed substantial computational and memory challenges in our setting, making it difficult to conduct experiments under our resource constraints.

---

### Official Review · Reviewer_b6Rj · 2025-10-27

**Soundness:** 2
**Presentation:** 2
**Contribution:** 2
**Rating:** 2
**Confidence:** 3

**Summary:**

This paper proposes the Laplace–Fourier Neural Operator (LFNO), which combines the Laplace Neural Operator (LNO) and Fourier Neural Operator (FNO) to capture both transient (decaying) and steady-state (periodic) dynamics. The Laplace branch models exponentially decaying modes via a pole–residue representation, while the Fourier branch handles oscillatory components through a Fourier integral operator. The authors evaluate LFNO on several ODEs and PDEs, claiming that it unifies the two representations and provides stable performance across both domains.

**Strengths:**

- The idea of bridging Laplace and Fourier domains is conceptually appealing and theoretically consistent.

- The implementation and dataset coverage are broad, spanning both transient ODEs and steady PDEs.

- The presentation and figures are clear, making the architecture easy to understand.

**Weaknesses:**

- Limited novelty: This combination seems a natural evolution after LNO, as Laplace and Fourier transforms are closely related. It is not striking as when FNO and LNO are just proposed. Therefore, meaningful novelty would only arise if LFNO demonstrated clear and strong outperformance or unique capabilities not achievable by FNO or LNO individually.

- Poor empirical performance on PDEs: In Table 3, LFNO consistently underperforms FNO across all PDEs, sometimes with up to 4× higher error. The text calls the results “comparable,” but this claim is not supported by the data. The lack of discussion on why LFNO performs worse or how to mitigate it is a major issue.

- Unclear theoretical framework: The integration of Laplace and Fourier components is not formally analyzed. It remains unclear why this hybridization should yield benefits, when each component dominates, or whether the Laplace branch introduces redundancy.

- Insufficient demonstration of value: The paper does not clearly identify specific regimes where LFNO clearly surpasses FNO or LNO for PDEs. The analysis could not specify where and why LFNO has advantages or should be considered.

- Potential confounding factors: The supposed stability advantage may stem from differences in hyperparameters, layer width, or training setup rather than from the Laplace–Fourier formulation itself. No ablation or sensitivity analyses are provided to isolate the cause.

Overall, LFNO presents a conceptually interesting bridge between Laplace and Fourier operator learning, but it is not convincingly demonstrated to work well in practice. The method underperforms FNO on PDEs, lacks theoretical clarity, and does not show strong justification for its additional complexity. The paper would benefit from a clearer theoretical grounding and from demonstrating specific problem types where this hybrid approach offers clear benefits.

**Questions:**

- In Table 3, LFNO shows higher L² errors than FNO on every PDE task, sometimes by factors of 2–4. Could you clarify the cause of this consistent degradation? Was this due to architectural constraints, optimization instability, or an inherent limitation of combining Laplace and Fourier components?

- Were FNO and LFNO both tuned equally for each PDE (e.g., mode count, width, learning rate)?

- Is there any class of problems where LFNO is essential?

---

> ### Author Response · Authors · 2025-12-03
> **Response to Reviewer b6Rj**
>
> We appreciate reviewer b6Rj for the detailed review and constructive questions. Our answers are as follows:
>
> 1. $\textbf{Limited Novelty and Unclear theoretical framework}$
>    - Existing neural operators have typically been designed to handle a single type of condition or regime within a given system. In contrast, we propose a hybrid architecture that first decomposes the solution into transient and steady-state components using LNO and subsequently applies a Fourier transform–based operator to the steady-state component. This establishes, for the first time, that a principled combination of Laplace-Fourier-based neural operators is not only feasible but also theoretically well-motivated.
>
> 2. $\textbf{Potential confounding factors and Were FNO and LFNO both tuned equally}$
>    - All models are equally tuned except for the gamma and activation function in PDEs experiments, since the given values made the best error rate for each experiment empirically. In the ODEs experiments, all models are equally tuned except for the activation functions. It could be confirmed in Table 4 in Appendix B.
>
> 3. $\textbf{Question 1. LFNO shows higher L² errors than FNO on every PDE task}$
>    - Our model is designed to approximate spectral dynamics, particularly in the presence of damping. We attempted to identify a PDE dataset that explicitly reflects this behavior, but were unable to find a suitable one. Consequently, our comparisons are based on the PDEs used in the LNO paper. Since these PDEs are primarily non-oscillatory or diffusive, our model—which excels at capturing spectral dynamics—tends to yield higher $\mathcal{L}_2$ errors compared to FNO in these settings. Our model produces the final prediction by adding the outputs of the transient layer and the steady-state layer. However, when training on PDEs that do not exhibit transient behavior, the transient layer does not diminish to zero as expected. Instead, it contributes residual values that effectively act as noise in the final output. We believe that this undesired contribution from the transient component is responsible for the elevated error rates in such cases. We emphasize that our model is not universally superior to FNO across all PDEs. Nevertheless, the key insight from these experiments is that our approach significantly outperforms LNO while still producing approximations that are close to those of FNO.

---

### Official Review · Reviewer_JavG · 2025-10-31

**Soundness:** 2
**Presentation:** 1
**Contribution:** 1
**Rating:** 2
**Confidence:** 3

**Summary:**

This paper introduces the Laplace–Fourier Neural Operator (LFNO), which combines the transient modeling capability of Laplace Neural Operators (LNO) with the steady-state modeling of Fourier Neural Operators (FNO). The approach is theoretically appealing and shows improved accuracy on selected damped and oscillatory systems.

**Strengths:**

LFNO unifies Laplace and Fourier Neural Operators, capturing both transient and steady-state dynamics. It improves accuracy and stability on damped and oscillatory systems, demonstrating superior performance over FNO and LNO on selected ODEs and PDEs.

**Weaknesses:**

* LFNO’s time–frequency hybrid representation is particularly suited for PDEs with damping, oscillations, or significant transient behavior. For purely stationary problems or short time windows, the added complexity may be unnecessary. The experiments are limited to a narrow class of PDEs, and it remains unclear whether LFNO offers advantages for undamped systems, strongly nonlinear chaotic systems, or PDEs with negligible transient dynamics. Clearer articulation of the applicable PDE classes is needed.

* The model expresses solutions as a sum of damped and oscillatory modes. However, in real data, damping and oscillation components may not be cleanly separable. It is unclear whether the learned “damped components” correspond to physical damping or merely capture generic time-decaying patterns. Quantitative validation is missing.

* Laplace transforms can be numerically unstable near poles in the complex plane, and gradient-based optimization in the Laplace domain may be sensitive. The paper does not provide experimental evidence that backpropagation in the Laplace domain is stable or that gradient flow remains well-behaved.

* LNO is not widely adopted as a practical baseline. A large variety of neural operators have been proposed recently, and intensive comparisons with these methods would be necessary to rigorously demonstrate LFNO’s advantages. Relevant recent works for comparison include:


  - L. Lei et al., U-WNO: U-Net enhanced wavelet neural operator for solving parametric partial differential equations, Comput. Math. Appl., 2025.


  - Z. Hao et al., GNOT: A General Neural Operator Transformer for Operator Learning, ICML 2023.


  - Z. Xiao et. al, Amortized Fourier Neural Operators, NeurIPS 2024.


  - RB. Rule et al., On the Benefits of Memory for Modeling Time-Dependent PDEs, ICLR 2025.

* Combining Laplace and Fourier transforms, pole–residue decomposition, and separate mode learning increases model complexity. While the paper claims efficiency improvements, no systematic analysis of training time, inference speed, memory usage, or parameter count is provided. Hyperparameter tuning requirements (e.g., number of modes or poles) may further limit practicality. Ablation studies are needed to clarify the contribution of each component.

**Questions:**

Please address the above weaknesses.

---

> ### Author Response · Authors · 2025-12-03
> **Response to Reviewer JavG**
>
> We thank reviewer JavG for the detailed review. Our responses are as follows:
>
> 1. $\textbf{Laplace transforms can be numerically unstable}$
>    - The theoretical foundation of LNO is established under the assumption that all poles are simple, and our model adheres to the same setting. This assumption allows the underlying function to be treated as a rational function, which provides a well-posed structure for the decomposition. Consequently, the method's stability is preserved.
>
> 2. $\textbf{Experiments are limited to a narrow class of PDEs}$
>    - The selection of experimental settings follows the scope established in the LNO literature, allowing for a direct and fair comparison under identical conditions. Furthermore, we first validate the proposed assumptions for PDEs that admit a well-defined decomposition structure, thereby ensuring that the methodology operates within our theoretical framework. Building on these results, we plan to extend the proposed conditions to more general PDEs in subsequent stages of the study.
>
> 3. $\textbf{More comparison for U-WNO, GNOT, Amortized FNO}$
>    - We appreciate this comment. We choose benchmarked algorithms to be as direct as possible to LFNO, as our first goal in empirical comparisons is to compare the three neural operators (LNO, FNO, LFNO) with closely related theoretical elements with minimal structural differences.
>
> 4. $\textbf{No systematic analysis of training time, inference speed, memory usage, or parameter count is provided}$
>    - We update parameter count, inference time, multiply-accumulate, and memory usage in Table 5 in Appendix B.

---

### Official Review · Reviewer_WWU7 · 2025-11-01

**Soundness:** 3
**Presentation:** 3
**Contribution:** 2
**Rating:** 4
**Confidence:** 4

**Summary:**

The authors propose Laplace–Fourier Neural Operator (LFNO), a framework that integrates LNO and FNO, where the transient response is captures by the LNO and the FNO the steady-state response. The authors claim this yields improved accuracy on systems with significant transient dynamics (e.g. undamped oscillators) while maintaining performance on standard PDE benchmarks. Empirical results are reported on three ODEs (Duffing oscillator, Lorenz system, pendulum) and five PDEs (Euler–Bernoulli beam, heat/diffusion equation, reaction–diffusion, Brusselator, Gray-Scott).

**Strengths:**

* The paper is well written, organized, and easy to follow.
* The problem of improved modelling of dynamical systems is of high importance to the ML community, and scientific community more broadly.
* Strong Empirical Performance on Transient Dynamics: LFNO clearly shines on problems with significant transient dynamics (decaying oscillations, initial condition effects). The results on the ODE benchmarks demonstrate substantial error reductions compared to both baseline models (the paper’s Table 3 shows much lower L2 errors for LFNO on undamped systems).
* Clear, principled decomposition: the transient/steady split is sensible and technically consistent with the underlying transforms; the architecture figure makes this easy to grasp.
* Training appears more stable than LNO on several tasks; breadth across 8 tasks (3 ODEs, 5 PDEs) shows generality within the tested setting.

**Weaknesses:**

* Empirical validation can be improved, as there are no error bars. I encourage the authors to run all experiments over multiple random seeds, and then report error bars, especially when making comparisons between methods.
* Limited Novelty (Essentially an Integration of Known Methods): The core idea of LFNO, while sensible, is ultimately a combination of two pre-existing operator learning techniques rather than an entirely new technique. LNO introduced using the Laplace domain for operator learning, and FNO is a well-established spectral operator model. This submission essentially places an FNO module on top of LNO’s steady-state output. The authors themselves note that LNO’s theory already “covers a broader scope” than FNO, and that the steady-state part of LNO naturally aligns with FNO’s mechanism. It lacks a unique theoretical contribution beyond what was in Cao et al. (2024) and Li et al. (2020) for FNO. This weakness is compounded by the empirical finding that on many tasks (especially the PDEs), LFNO performs on par with standard FNO, not surpassing it. That means the combined model’s advantage manifests mainly in scenarios that LNO already targeted (transients in time), and it doesn’t push beyond FNO for the broader class of problems. For a paper claiming a generally “effective operator learning” solution, not exceeding FNO on steady-state problems is a bit underwhelming in terms of novelty and impact.
* Missing Comparisons to Other State-of-the-Art Methods: The experiments compare LFNO only against two baselines: FNO and LNO. Interestingly the authors cite Factorized FNO (F-FNO) by Tran et al. 2023 and Geo-FNO by Li et al. 2023 for general geometries, yet do not compare against them.
* Empirical Scope is Narrow (No Large-Scale or 2D/3D Experiments): All experiments in the paper involve either ODEs or 1D spatial PDEs (with time). The spatial domain in PDE cases is one-dimensional with a 50-point grid. This doesn’t satisifactorly answer the question of how well would LFNO scale to more complex domains or higher dimensions?
* Missing baselines that specifically target time/localization or non-Fourier spectra: please compare against TNO (Temporal Neural Operator), PDNO / Spectral Neural Operator, and Wavelet Neural Operator; and include at least one attention/transformer-based operator (e.g., Galerkin Transformer / Operator-Transformer / GNOT) to substantiate broader “effective operator learning” claims.
* No complexity/efficiency analysis. Parameter counts, training time, and inference cost vs. FNO/LNO are not reported. If LFNO is strictly more complex, clarify when its extra cost is justified.
* Lack of ablations. It would help to separate “combination benefit” from “capacity benefit”: e.g., deeper LNO vs. LFNO; FNO augmented with a simple decay channel vs. LFNO; varying the number of transient/steady layers.

**Questions:**

* Comparison with Other Baselines: How does LFNO compare with Temporal Neural Operator (TNO) or other recent neural operators on time-dependent problems?
* Applicability to Higher Dimensions / General Geometries: Since your steady-state layer currently uses Fourier transforms, how would LFNO handle a case with a non-uniform domain or 2D/3D spatial grids where FFT is not directly applicable?
* Complexity: Please report parameter counts, wall-clock training time, and inference throughput vs. FNO and LNO on at least one representative task.
* Ablation: What performance do you get as you vary the transient/steady layer counts (e.g., 2:2, 4:1, 1:4)? Does a deeper LNO close the gap on ODEs?

---

> ### Author Response · Authors · 2025-12-03
> **Response to Reviewer WWU7**
>
> We appreciate reviewer WWU7 for the detailed review and valuable questions. Our responses are as follows:
>
> 1. $\textbf{Limit Novelty (Essentially an Integration of Known Methods)}$
>    - Existing neural operators have typically been designed to handle a single type of condition or regime within a given system. In contrast, we propose a hybrid architecture that first decomposes the solution into transient and steady-state components using LNO and subsequently applies a Fourier transform–based operator to the steady-state component. This establishes, for the first time, that a principled combination of Laplace-Fourier-based neural operators is not only feasible but also theoretically well-motivated.
>
> 2. $\textbf{Comparison with other baselines}$
>    - We appreciate this comment. We choose benchmarked algorithms to be as direct as possible to LFNO, as our first goal in empirical comparisons is to compare the three neural operators (LNO, FNO, LFNO) with closely related theoretical elements with minimal structural differences.
>
> 3. $\textbf{Complexity}$
>    - We update parameter count, inference time, multiply-accumulate, and memory usage in Table 5 in Appendix B.
>
> 4. $\textbf{Applicability to Higher Dimensions / General Geometries}$
>    - We note the reviewer’s concern regarding non-uniform domains and higher-dimensional grids. LFNO can handle such cases, as seen in the approach used in the Fourier Neural Operator. Although this is not explicitly illustrated in the architecture figure, each layer employs a linear transformation W that allows the model to capture non-periodic boundary conditions, thereby extending applicability to non-uniform domains and higher-dimensional spatial grids.
>
> 5. $\textbf{Ablation Study}$
>    - Thank you for highlighting this issue. We have already conducted experiments adjusting the number of transient layers and steady layers, and updated Table 6 in Appendix C. When the number of transient layers is fixed, increasing the number of steady layers initially shows a sharp difference. Still, this difference becomes negligible when the number of layers increases from 3 to 4. When the number of steady layers is fixed, increasing the number of transient layers does not show a significant difference.

---

### Meta-Review · Area_Chair_oba1 · 2026-01-06

**Summary:**

* Strength: The paper provides a framework that integrates Laplace and Fourier transforms to model transient and steady-state dynamics
* Strength: The proposed architecture shows better training stability and error reduction on damped ODE systems compared to the original Laplace Neural Operator.
* Weakness: The model show a significant performance regression on steady-state PDE benchmarks: it produces much higher errors than the standard Fourier Neural Operator.
* Weakness: The experimental evaluation lacks statistical rigor: does not include error bars or results from multiple random seeds.
* Weakness: The datasets used are small and synthetic.
* Weakness: The authors admit the transient branch introduces noise into the steady-state solution

The Laplace-Fourier Neural Operator (LFNO) uses a pole-residue decomposition to isolate transient components and applies a spectral kernel to the remaining signal. This theoretical approach is well-motivated and addresses known limitations in how existing models handle damping effects in dynamical systems.

Results on simple ODEs are positive but the paper's claims of being comparable to FNO on PDEs are not supported by the data. The reliance on very small datasets (380 samples) + absence of error bars make it difficult to verify the reliability. The model currently lacks a mechanism to prevent the transient branch from degrading accuracy when no transient behavior is present.

**Reviewer Concerns:**

**Note:** there was no time for the reviewers to engage in discussion during the rebuttal period.

**Addressed by rebuttal**
* [Non-core] Lack of information regarding model complexity and efficiency -- see new Table 5 in Appendix B.
* [Non-core] Missing sensitivity analysis for the number of layers in each branch -- see Table 6 in Appendix C.
* [Non-core] Clarification on how the model handles non-uniform domains -- see text explanation (but reviewers could not comment)

**Still outstanding**
* [Core] Significant accuracy gap compared to the Fourier Neural Operator on standard PDEs.
* [Core] Lack of error bars and statistical significance for the results.
* [Core] Missing comparisons to modern state-of-the-art temporal operators like TNO.
* [Core] Numerical stability of the pole-residue decomposition in the Laplace domain.

The authors provided helpful clarifications on model complexity and layer sensitivity. However, the most critical technical issue remains. The model is less accurate than existing baselines on standard problems because the transient branch interferes with the steady-state prediction. This issue, combined with the lack of error bars and small dataset sizes, means the primary claims of the paper are not yet sufficiently validated.

**Reviewer Scores:**

* Reviewer ID: WWU7
* Original score: 4
* Estimated score shift: unchanged

* Reviewer ID: JavG
* Original score: 2
* Estimated score shift: unchanged
* Justification: The reviewer correctly identified fundamental issues regarding mode separation and numerical stability & this was only partially addressed by the authors.

* Reviewer ID: b6Rj
* Original score: 2
* Estimated score shift: unchanged
* Justification: The reviewer accurately pointed out the significant performance regression on PDEs: this contradicts the paper's claim of being comparable to FNO.

* Reviewer ID: NutN
* Original score: 4
* Estimated score shift: decrease
* Justification: This reviewer was overly optimistic about the results; the score would likely drop once the significant error gap on PDEs is factored into the assessment.

The consensus seems to be that the paper has a clear motivation but falls short in its empirical execution. Reviewers b6Rj and JavG correctly focused on the performance degradation and stability risks. Reviewers WWU7 and NutN were more positive but still flagged the incremental nature + lack of rigor.

---

### Decision · Program_Chairs · 2026-01-26

Reject